# Time after ostomy surgery and type of treatment are associated with quality of life changes in colorectal cancer patients with colostomy

Karine de Almeida Silva[1], Arenamoline Xavier Duarte[1‡], Amanda Rodrigues Cruz[1‡], Lúcio Borges de Araújo[2‡], Geórgia das Graças Pena[1,3]*

1 Graduate Program in Health Sciences, Federal University of Uberlandia, Uberlandia, Minas Gerais, Brazil, 2 School of Mathematics, Federal University of Uberlandia, Uberlandia, Minas Gerais, Brazil, 3 School of Medicine, Nutrition Course, Federal University of Uberlandia, Uberlandia, Minas Gerais, Brazil

☯ These authors contributed equally to this work.
‡ These authors also contributed equally to this work.
* georgia@ufu.br, georgiapena@gmail.com

**Data Availability Statement:** All relevant data are within the manuscript and its Supporting Information files.

## Abstract

### Purpose

Quality of life in colorectal cancer patients may be affected by colostomy and treatment, but relevant studies are still scarce and contradictory. The present study aimed to evaluate the association between colostomy time and treatment type with quality of life in colorectal cancer patients.

### Methods

A prospective observational study of 41 patients with colorectal cancer was conducted on three occasions T0, T1 and T2 (0–2; 3–5 and 6–8 months after ostomy surgery, respectively). The treatments prescribed were: surgery alone, chemotherapy or radiotherapy, or chemoradiotherapy. European Organization for Research and Treatment of Cancer questionnaires were used to evaluate quality of life. Worsening clinical changes were evaluated considering difference in scores between times of surgery $\geq \pm 9$ points.

### Results

Regarding ostomy surgery, scores in physical function improved between T0 and T1 and these better scores were maintained at T1 to T2. The same was observed for urinary frequency, appetite loss and dry mouth. Chemoradiotherapy was associated with worse scores for global health status, nausea and vomiting, bloating and dry mouth. Although significant differences were not observed in some domains in the Generalized Estimating Equations analysis, patients showed noticeable changes for the worse in the pain, anxiety, weight concern, flatulence and embarrassment domains during these periods.

**Funding:** The Coordination of Improvement of Higher Education Personnel (Coordenação de Aperfeiçoamento de Pessoal de Nível Superior - CAPES) and the National Council for Scientific and Technological Development (Conselho Nacional de Desenvolvimento Científico e Tecnológico- CNPq), Brazil funded the publication fee. The funders had no role in study design, data collection and analysis, decision to publish, or preparation of the manuscript.

**Competing interests:** The authors declare that they have no conflict of interest.

## Conclusions

Colostomy improved quality of life at 3–5 months in most domains of quality of life and remained better at 6–8 months after surgery. Chemoradiotherapy had a late negative influence on quality of life. Health teams could use these results to reassure patients that this procedure will improve their quality of life in many functional and symptomatic aspects.

## Introduction

Colorectal cancer (CRC) has a high prevalence worldwide. According to global cancer statistics, new CRC cases in 2018–2020 were estimated at almost 2 million with more than 900,000 deaths, making it the third most frequently diagnosed cancer in males and the second in females [1,2].

The treatment of this cancer involves relieving symptoms (mainly low digestive haemorrhage, abdominal mass, abdominal pain, change in bowel habits, weight loss and anemia), limiting or ceasing disease progression; it may require chemo/radiotherapy treatment, bowel resection and ostomy confection [3,4]. The frequency of ostomy for CRC varies widely, from 6 to 47% of colostomies and from 5 to 69% of ileostomies, depending on the study design and population [5–8]. In addition, the presence of this cancer could lead to functional limitations, cognitive changes and emotional stress affecting the patient's general quality of life (QoL) [9]. However, studies regarding the impact of ostomy and type of treatment on QoL of patients who have had colostomy for CRC are still scarce and have contradictory results. While some studies showed ostomy to be associated with improved QoL [10], it was associated with worse QoL in others [11,12] and one study found no difference [13].

Work-life and productivity, interpersonal relationships and other social activities can be impacted by treatment [14] and ostomy surgery [11,12]. For example, it is common for patients to present a worsening of physical function, cognitive, role and social function and symptoms of pain, nausea and vomiting and constipation due to chemotherapy. Radiotherapy can cause problems in sexual and urinary function [15]. Ostomy surgery can also affect the QoL of these patients because there may be physical and psychological difficulties related to the limiting aspects of the stoma, such as activities of daily living, work capacity, and social interaction [16].

Although some studies have evaluated the QoL of patients with ostomy for CCR. However, very few of these were prospective studies [6,12,17,18]. Most were case-control [13,19,20] or cross-sectional [14,21,22] in design or compared patients with and without ostomies [5,6,10–12,23–25]. Some evaluated different reasons for ostomies [26], or considered colostomy and ileostomy together [8,17]. The few prospective studies developed did not evaluate QoL with a specific questionnaire for cancer patients [17,18]. So, to the best of our knowledge, this is the first prospective study to evaluate the association of ostomy surgery time and type of treatment in a group of specifically CRC patients, using an appropriate QoL questionnaire for cancer patients. This perspective is important since the symptoms and type of treatment are different among ostomized patients (ileum or colon) depending on the disease (e.g. cancer or inflammatory bowel disease). So, QoL may be affected differently when only colostomies are evaluated and only for one disease, i.e. CRC. In addition, some studies considered chemotherapy and radiation treatment and others did not.

These differences in studies design and population mix may explain the contradictory results between the authors and may not clearly show the real impact of the ostomy on

individuals over time. A prospective study with a specific sample may clarify these outcomes, as it would allow evaluation over time. Therefore, the present study aimed to evaluate the association between colostomy time and treatment type with QoL in CRC patients. We hypothesized that ostomy surgery and chemoradiotherapy could negatively change QoL in this follow-up.

## Methods

### Participants and procedures

**Design of study and ethical aspects.** A prospective study was performed between August 2017 and May 2019 in a university hospital, among both hospitalized patients and outpatients, who had been given either a temporary or permanent colostomy for CRC a maximum of two months previously.

The Human Research Ethics Committee approved this study (protocol number 65975817.6.0000.5152), and all participants signed a free and informed consent form.

**Inclusion and exclusion criteria.** The study included all patients aged 18 years or older given a colostomy for CRC up to 2 months previously, regardless of stage, including those receiving palliative treatment. The following patients were excluded from the study: those with previous diagnosis of major depression, neuropsychopathies or other serious mental illness registered in clinical records, those with chronic diseases that required intense food intake modification and those who underwent bowel transit reconstruction during follow-up.

**Data collection.** Three interviews were conducted: T0, T1 and T2: T0 (0–2 months postoperative), T1 (3 months after T0) and T2 (3 months after T1). Thus, T1 was 3 to 5 months and T2 6 to 8 months after ostomy surgery (Fig 1). Overall, in the study period, 41 patients participated at baseline, 15 in T1 and 16 in T2 at the end (Fig 1).

Diagnosis, ostomy surgery, intestinal resection, anatomopathological, treatment type and other clinical data were collected from clinical records. Sociodemographic (at T0), anthropometric and QoL data were collected by face to face interview on three occasions.

**Treatment types.** The standard treatment protocol in the evaluated service is neoadjuvant chemoradiotherapy (5 weekly cycles of 5-fluorouracil (5-FU) at 500mg / $m^2$ for chemotherapy and 5 cycles of 50 Gy radiotherapy concurrently with chemotherapy) for patients with rectal cancer. The protocol for patients diagnosed with colon cancer is primarily surgery and in advanced cases, adjuvant chemotherapy. Some patients receive treatment only after ostomy surgery, if the diagnosis was made during the bowel resection procedure. Also, if complete tumor resection cannot be achieved, patients undergo adjuvant chemotherapy.

In the present study, investigating the different types and moments of treatment resulted in a small number of individuals, making it necessary to group them for a better analysis. The treatment types were grouped, regardless of treatment period: surgery alone (S), surgery plus chemotherapy or surgery plus radiotherapy (CT/RT), and surgery plus chemoradiotherapy (CRT).

### Measures

**Anthropometrics measures.** Weight and height were measured at T0, T1 and T2, according to the World Health Organization (WHO) protocols [27]. Body mass index (BMI) was calculated from body weight (kg) divided by the square of height (m), following WHO reference for adults under 59 years old [kg/$m^2$ (< 18.5 malnourished; ≥18.5 to < 25 well-nourished; ≥ 25 to < 30 overweight; and ≥ 30 obesity)], and the Pan American Health Organization for those aged 60 years or more [kg/$m^2$ (< 23.0 malnourished; ≥ 23 to < 28 well-nourished; ≥ 28 to < 30 overweight; and ≥ 30 obesity)], [28–30].

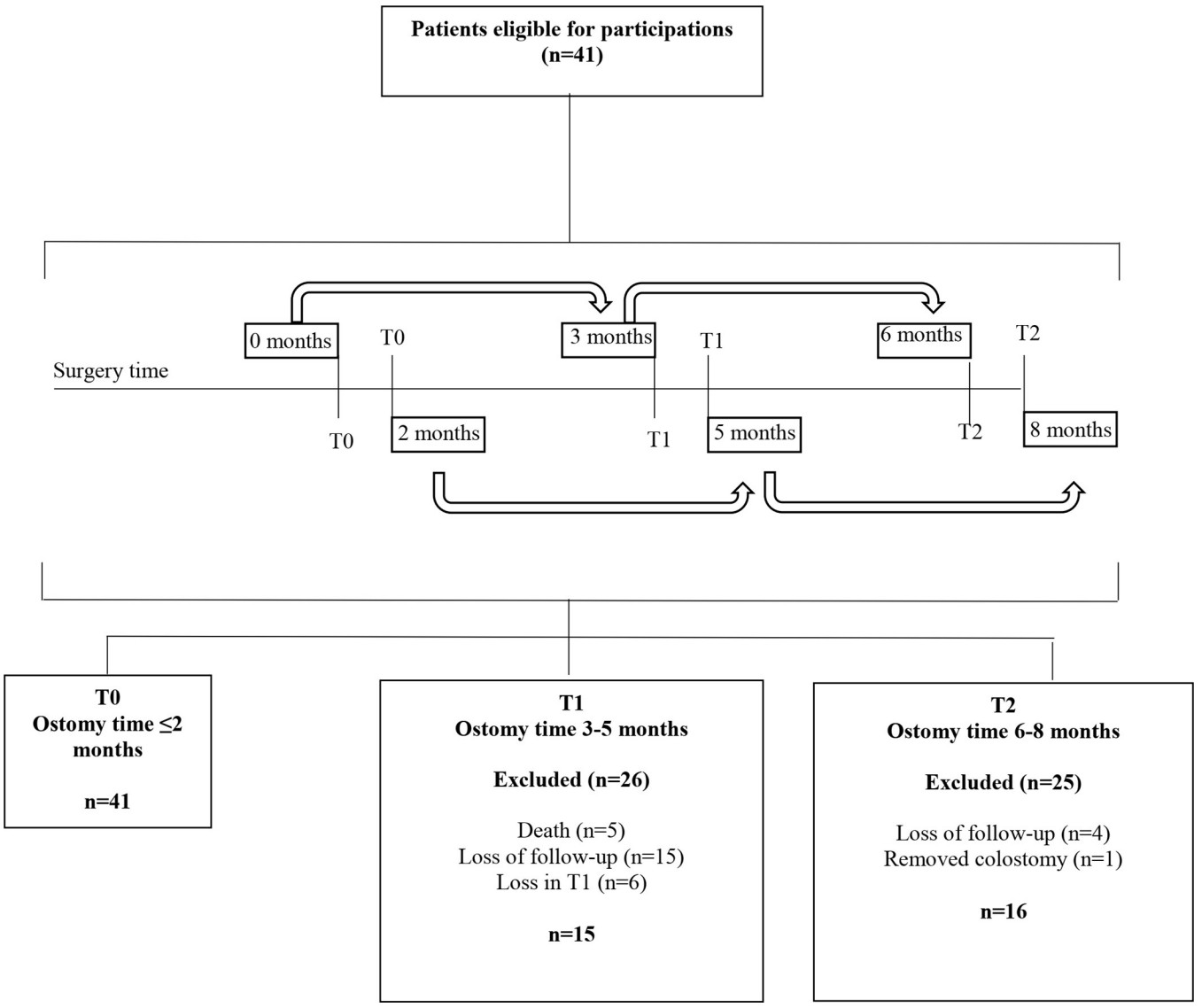

**Fig 1. Diagram reporting the number of patients with ostomies due to colorectal cancer screened and recruited in this study.**

**Quality of life questionnaires.** European Organization for Research and Treatment of Cancer questionnaires, EORTC-QLQ-C30 version 3.0 and EORTC-QLQ-CR29, were used to assess QoL, after authorization. EORTC-QLQ-C30 version 3.0 is a questionnaire for all cancer patients and consists of 30 questions, which are divided into scales: 5 for function and 2 for QoL (physical, emotional, cognitive, social, role performance, overall health and QoL); 3 for symptoms (fatigue, pain and nausea and vomiting); and 6 unique items (symptoms and financial impact of the disease). EORTC-QLQ-CR29 is a questionnaire for CRC patients containing 29 questions comprising 4 scales (urinary frequency, blood and mucus in stool, stool frequency and body image) and 19 unique items. According to the authors, this questionnaire should always be applied with EORTC-QLQ-C30 [31]. All scales and single items measures in both questionnaires were transformed to estimate scores from 0 to 100 according to the algorithm recommended by the EORTC scoring guidelines. For global QoL and function scales, a higher

score means better function, and better QoL; for symptom scales, a higher scale means higher symptom burden and worse QoL [31,32].

## Statistical analysis

The distribution of variables was tested by Kolmogorov-Smirnov test. Descriptive statistics were expressed as number and percentage or mean (standard deviation) or median (interval interquartile). Friedman's non-parametric test was used to test the differences in QoL scores between times.

Since all patients given colostomy surgery for CCR during the collection period were invited to participate, we did not perform sample size calculation a priori, but the post hoc effect size and observation power were performed a posteriori. All effect sizes for each domain were calculated by the Cohen method [33]. A post hoc test was then performed to estimate the observation power using the Generalized Estimating Equations (GEE) test, 0.05 error, sample size 41 at T0, 15 at T1 and 16 at T2, using R version 3.5.3 [34]. Thus, the observation power of each domain was obtained. For the results and discussion of the present study, we considered only those domains with power equal to or above 0.80 in the prospective time and treatment type analysis. To account for losses during follow-up, we calculated baseline differences between the group that remained in the study and the group that did not complete using the ANOVA, Chi-square or Mann-Whitney tests (S1–S3 Tables).

The GEE was used to estimate the association between time of colostomy (T0, T1 and T2) and type of treatment (S, CT/RT and CRT) considering the main exposures and the QoL domains (endpoints). The GEE models provide consistent estimates of the criteria for standard errors through robust estimators correlating the within-subject outcome variables of the treatment performed and the time of colostomy. The lower quasi-likelihood under the independence model criterion (QIC) was tested for all models, and the Tweedie with log link model was used in GEE analysis. The Bonferroni post hoc test was used for multiple comparisons and sex, age (years), BMI (kg/m$^2$) and disease stage were considered as potential cofounders.

Although statistical differences are the final goal in many studies, some differences could be considered clinically important. As a result, recent studies with EORTC [35,36] recommended the evaluation of between score changes during follow-up in order to make a comprehensive assessment of QoL aspects. Following this recomendation, the cutoff value for our population was estimated by effect size (0.5) x standard deviation of the study population–considering global QoL score (SD = 18.45) = 9.23. Thus, difference in scores between times ≥±9 points in the scales by time was considered as clinically important, as in previous studies [37,38]. Only worsening between times was shown, holding the worse direction of the scales and items. The analyses were performed with SPSS version 24 for Windows (SPSS Inc., Chicago, Illinois, United States of America) and p-value ≤0.05 was considered statistically significant.

## Results

Sociodemographics and clinical data are shown in Table 1. A total of 41 patients were included in this study; 53.7% were female, 43.9% were between 60 and 70 years old and 29.3% had an income of less than the two monthly minimum wage. Chemoradiotherapy was given to 51.2% of patients. Regarding nutritional diagnosis, 22.5% were malnourished at T0 which reduced to 14.3% at T2. At T0, 25.0% of patients were overweight or obese which increased to 33.3% at T2.

Regarding the EORTC-QLQ-C30 and EORTC-QLQ-CR29 scores at different time periods, some differences were found in physical function, with the worst score at T0 (Table 2). Some symptom scales were also worse at T0 including fatigue, bloating, dry mouth and stoma care

**Table 1. Sociodemographic and clinical data of patients with colostomy for colorectal cancer (n = 41).**

| Variables | | % (n) |
|---|---|---|
| **Sociodemographic and economic** | | |
| **Age (years)** | | |
| < 60 | | 36.6 (15) |
| ≥ 60 to <65 | | 24.4 (10) |
| ≥65 to < 70 | | 19.5 (8) |
| ≥ 70 | | 19.5 (8) |
| **Gender** | | |
| Male | | 46.3 (19) |
| Female | | 53.7 (22) |
| **Monthly Minimum Wage***  | | |
| < 1 | | 7.3 (3) |
| ≥ 1 to < 2 | | 22.0 (9) |
| ≥ 2 to < 3 | | 34.1 (14) |
| ≥ 3 | | 29.3 (12) |
| Not specified | | 7.3 (3) |
| **Clinical Diagnosis** | | |
| Colon tumor | | 34.1 (14) |
| Rectal tumor | | 65.9 (27) |
| **Staging** | | |
| I | | 22.0 (9) |
| II | | 19.5 (8) |
| III | | 29.3 (12) |
| IV | | 9.8 (4) |
| Pathological staging Y | | 13.0 (6) |
| Not specified or unknown | | 4.9 (2) |
| **Treatment** | | |
| Surgery only | | 14.6 (6) |
| Chemotherapy or radiotherapy | | 34.1 (14) |
| Chemoradiotherapy | | 51.2 (21) |
| **Intention of primary treatment** | | |
| Curative | | 68.3 (28) |
| Palliative | | 31.7 (13) |
| **Ostomy** | | |
| Permanent | | 29.3 (12) |
| Temporary | | 70.7 (29) |
| **Comorbidities** | | |
| Diabetes | | 14.6 (6) |
| Systemic Arterial Hypertension | | 36.6 (15) |
| Heart diseases | | 12.2 (5) |
| Other | | 17.1 (7) |
| **Death** | | 12.2 (5) |
| **Nutritional status** | | |
| **T0** | Malnourished | 22.5 (9) |
| | Overweight / Obesity | 25.0 (10) |
| **T1** | Malnourished | 25.0 (6) |
| | Overweight / Obesity | 25.0 (6) |
| **T2** | Malnourished | 14.3 (3) |

(*Continued*)

**Table 1.** (Continued)

| Variables | % (n) |
|---|---|
| Overweight / Obesity | 33.3 (7) |

*1 Monthly Minimum Wage was equivalent to US$312.00

problems (Tables 2 and 3). The scores for these domains improved at T1 and this was maintained at T2.

In the GEE analysis, many QoL domains improved between T0 and T1 and this improvement was maintained between T1 and T2. These included physical function, appetite loss, urinary frequency and dry mouth, showing a mean difference range from 10 to 49 points in the scores between times (Tables 4 and 5). The emotional function scale showed worsening at T1 to T2 while the insomnia domain improved at T1 to T2 and nausea and vomiting, insomnia,

**Table 2. Synopsis of EORTC QLQ-C30 longitudinal score differences and corresponding p-values according to quality of life scales for patients with colostomy due to colorectal cancer.**

| Quality of Life domains | T0 | | T1 | | T2 | | p-value | Observation Power T0-T1 | Observation Power T1-T2 | Observation Power T0-T2 |
|---|---|---|---|---|---|---|---|---|---|---|
| | Mean (SD) | Median (p25-p75) [min-max] | Mean (SD) | Median (p25-p75) [min-max] | Mean (SD) | Median (p25-p75) [min-max] | | | | |
| Global health status | 74.16 (20.95) | 79.16 (66.66–85.41) | 82.50 (18.61) | 91.66 (64.58–100.00) | 80.83 (20.05) | 87.50 (64.58–100.00) | 0.157 | **0.90** | **0.99** | **0.87** |
| Physical function | 78.00 (14.53) | 80.00 (65.00–85.75) | 94.50 (9.26) | 97.50 (93.75–100.00) | 95.00 (8.16) | 100.00 (90.00–100.00) | **0.005** | **0.99** | 0.07 | **0.96** |
| Role function | 68.33 (36.38) | 75.00 (41.66–100.00) | 78.33 (34.29) | 100.00 (50.00–100.00) | 81.66 (25.39) | 100.00 (62.50–100.00) | 0.163 | **0.96** | 0.78 | 0.64 |
| Emotional function | 57.50 (37.15) | 62.50 (20.83–100.00) | 71.66 (29.44) | 75.00 (62.50–93.75) | 73.33 (31.13) | 79.16 (56.25–100.00) | 0.908 | 0.37 | **0.99** | 0.62 |
| Cognitive function | 73.33 (25.09) | 66.66 (50.00–100.00) | 76.66 (29.60) | 83.33 (66.66–100.00) | 80.00 (20.48) | 83.33 (66.66–100.00) | 0.772 | **0.91** | **0.99** | 0.43 |
| Social function | 80.00 (33.14) | 100.00 (62.50–100.00) | 95.00 (11.24) | 100.00 (95.83–100.00) | 95.00 (11.24) | 100.00 (95.83–100.00) | 0.086 | 0.39 | 0.33 | 0.08 |
| Fatigue | 42.22 (35.05) | 33.33 (19.44–75.00) | 15.55 (31.51) | 0.00 (0.00–16.66) | 10.00 (24.81) | 0.00 (0.00–5.55) | **0.012** | **0.96** | 0.40 | 0.18 |
| Nausea and vomiting | 18.33 (33.74) | 0.00 (0.00–37.50) | 8.33 (18.00) | 0.00 (0.00–8.33) | 6.66 (21.08) | 0.00 (0.00–0.00) | 0.779 | 0.14 | **0.98** | **0.85** |
| Pain | 25.00 (36.21) | 0.00 (0.00–54.16) | 15.00 (19.95) | 8.33 (0.00–25.00) | 18.33 (32.82) | 0.00 (0.00–25.00) | 0.962 | **0.99** | 0.45 | 0.05 |
| Dyspnea | 6.66 (21.08) | 0.00 (0.00–0.00) | 0.00 (0.00) | 0.00 (0.00–0.00) | 0.00 (0.00) | 0.00 (0.00–0.00) | 0.368 | 0.06 | **0.99** | 0.05 |
| Insomnia | 36.66 (42.88) | 16.66 (0.00–75.00) | 36.66 (45.67) | 16.66 (0.00–100.00) | 13.33 (32.20) | 0.00 (0.00–8.33) | 0.325 | **0.99** | **0.93** | **0.99** |
| Appetite loss | 16.66 (36.00) | 0.00 (0.00–16.66) | 10.00 (31.62) | 0.00 (0.00–0.00) | 3.33 (10.54) | 0.00 (0.00–0.00) | 0.223 | **0.99** | **0.99** | 0.53 |
| Constipation | 20.00 (43.16) | 0.00 (0.00–25.00) | 3.33 (10.54) | 0.00 (0.00–0.00) | 0.00 (0.00) | 0.00 (0.00–0.00) | 0.156 | 0.57 | **0.99** | 0.66 |
| Diarrhea | 23.33 (41.72) | 0.00 (0.00–50.00) | 3.33 (10.54) | 0.00 (0.00–0.00) | 10.00 (31.62) | 0.00 (0.00–0.00) | 0.368 | 0.50 | 0.05 | **0.99** |
| Financial difficulties | 16.66 (36.00) | 0.00 (0.00–16.66) | 20.00 (42.16) | 0.00 (0.00–25.00) | 3.33 (10.54) | 0.00 (0.00–0.00) | 0.779 | 0.07 | **0.99** | 0.15 |

Data presented for mean, standard deviation, median, interquartile range or minimum and maximum. p <0.05 was considered significant (Friedman test).

**Table 3. Synopsis of EORTC QLQ-CR29 longitudinal score differences and corresponding p-values according to quality of life scales for patients with colostomy due to colorectal cancer.**

| Quality of Life domains | T0 | | T1 | | T2 | | p-value | Observation Power T0-T1 | Observation Power T1-T2 | Observation Power T0-T2 |
|---|---|---|---|---|---|---|---|---|---|---|
| | Mean (SD) | Median (p25-p75) [min-max] | Mean (SD) | Median (p25-p75) [min-max] | Mean (SD) | Median (p25-p75) [min-max] | | | | |
| Urinary frequency | 8.33 (14.16) | 0.00 (0.00–20.83) | 0.00 (0.00) | 0.00 (0.00–0.00) | 13.33 (28.10) | 0.00 (0.00–16.66) | 0.178 | **0.95** | 0.14 | **0.95** |
| Blood or mucus in stools | 0.00 (0.00) | 0.00 (0.00–0.00) | 0.00 (0.00) | 0.00 (0.00–0.00) | 3.33 (7.02) | 0.00 (0.00–4.16) | 0.135 | **0.99** | 0.20 | 0.71 |
| Stool frequency | 8.33 (26.35) | 0.00 (0.00–0.00) | 3.33 (10.54) | 0.00 (0.00–0.00) | 0.00 (0.00) | 0.00 (0.00–0.00) | 0.607 | **0.99** | 0.21 | 0.52 |
| Body image | 82.22 (35.60) | 100.00 (75.00–100.00) | 64.44 (43.12) | 88.88 (25.00–100.00) | 72.22 (36.38) | 88.88 (47.22–100.00) | 0.101 | 0.45 | 0.05 | **0.99** |
| Urinary incontinence | 3.33 (10.54) | 0.00 (0.00–0.00) | 10.00 (22.49) | 0.00 (0.00–8.33) | 0.00 (0.00) | 0.00 (0.00–0.00) | 0.368 | 0.11 | 0.05 | **0.81** |
| Dysuria | 26.66 (40.97) | 0.00 (0.00–50.00) | 6.66 (21.08) | 0.00 (0.00–0.00) | 10.00 (31.62) | 0.00 (0.00–0.00) | 0.223 | 0.34 | **0.99** | 0.77 |
| Abdominal pain | 26.66 (37.84) | 0.00 (0.00–66.66) | 3.33 (10.54) | 0.00 (0.00–0.00) | 16.66 (36.00) | 0.00 (0.00–16.66) | 0.431 | **0.99** | 0.13 | 0.17 |
| Buttock pain | 20.00 (35.83) | 0.00 (0.00–41.66) | 26.66 (40.97) | 0.00 (0.00–50.00) | 13.33 (32.20) | 0.00 (0.00–8.33) | 0.273 | **0.99** | 0.37 | **0.99** |
| Bloating | 30.00 (39.90) | 16.66 (0.00–50.00) | 3.33 (10.54) | 0.00 (0.00–0.00) | 0.00 (0.00) | 0.00 (0.00–0.00) | **0.030** | 0.53 | **0.82** | 0.05 |
| Dry mouth | 60.00 (34.42) | 66.66 (33.33–100.00) | 16.66 (32.39) | 0.00 (0.00–33.33) | 13.33 (23.30) | 0.00 (0.00–33.33) | **0.018** | **0.99** | **0.89** | **0.99** |
| Hair loss | 20.00 (35.83) | 0.00 (0.00–41.66) | 13.33 (32.20) | 0.00 (0.00–8.33) | 10.00 (16.10) | 0.00 (0.00–33.33) | 0.926 | **0.99** | 0.10 | 0.48 |
| Taste | 10.00 (22.49) | 0.00 (0.00–8.33) | 0.00 (0.00) | 0.00 (0.00–0.00) | 10.00 (31.62) | 0.00 (0.00–0.00) | 0.368 | 0.71 | 0.28 | 0.38 |
| Anxiety | 36.66 (48.30) | 0.00 (0.00–100.00) | 63.33 (36.68) | 66.66 (33.33–100.00) | 50.00 (45.13) | 66.66 (0.00–100.00) | 0.156 | 0.32 | 0.12 | **0.99** |
| Weight | 80.00 (35.83) | 100.00 (58.33–100.00) | 66.66 (35.13) | 66.66 (33.33–100.00) | 80.00 (32.20) | 100.00 (66.66–100.00) | 0.309 | **0.99** | 0.16 | 0.06 |
| Flatulence | 26.66 (40.97) | 0.00 (0.00–50.00) | 26.66 (40.97) | 0.00 (0.00–50.00) | 23.33 (31.62) | 16.66 (0.00–33.33) | 0.956 | 0.05 | 0.05 | 0.66 |
| Fecal incontinence | 10.00 (31.62) | 0.00 (0.00–0.00) | 3.33 (10.54) | 0.00 (0.00–0.00) | 10.00 (16.10) | 0.00 (0.00–33.33) | 0.368 | 0.36 | 0.26 | 0.50 |
| Sore skin | 10.00 (31.62) | 0.00 (0.00–0.00) | 10.00 (22.49) | 0.00 (0.00–8.33) | 16.00 (23.57) | 0.00 (0.00–33.33) | 0.449 | **0.80** | 0.04 | **0.99** |
| Embarrassment | 33.33 (41.57) | 16.66 (0.00–75.00) | 20.00 (32.20) | 0.00 (0.00–33.33) | 30.00 (39.90) | 16.66 (0.00–50.00) | 0.513 | 0.27 | 0.40 | **0.99** |
| Stoma care problems | 46.66 (50.18) | 33.33 (0.00–100.00) | 10.00 (31.62) | 0.00 (0.00–0.00) | 0.00 (0.00) | 0.00 (0.00–0.00) | **0.015** | 0.57 | 0.21 | **0.99** |
| Sexual interest (men) | 16.66 (33.33) | 0.00 (0.00–50.00) | 16.66 (19.24) | 16.66 (0.00–33.33) | 66.66 (38.49) | 66.66 (33.33–100.00) | 0.323 | **0.99** | 0.06 | **0.99** |
| Impotence | 16.66 (33.33) | 0.00 (0.00–50.00) | 8.33 (16.66) | 0.00 (0.00–25.00) | 25.00 (31.91) | 16.66 (0.00–58.33) | 0.607 | 0.41 | **0.99** | 0.13 |
| Sexual interest (women) | 11.11 (27.21) | 0.00 (0.00–16.66) | 22.22 (40.36) | 0.00 (0.00–50.00) | 16.67 (40.82) | 0.00 (0.00–25.00) | 0.867 | 0.68 | **0.99** | **0.99** |
| Dyspareunia | 0.00 (0.00) | 0.00 (0.00–0.00) | 0.00 (0.00) | 0.00 (0.00–0.00) | 22.22 (40.36) | 0.00 (0.00–50.00) | 0.135 | **0.99** | 0.07 | 0.55 |

Data presented for mean, standard deviation, median, interquartile range or minimum and maximum. p <0.05 was considered significant (Friedman test).

**Table 4. Model effect, comparisons and post hoc tests of the variables of quality of life of EORTC QLQ-C30 with time and treatment factors using Generalized Estimating Equations.**

| Quality of Life domains | Effect | Df[3] | Wald χ2 | p-value | Comparisons (pairwise method) | Mean difference (I-J) | p–value (Bonferroni) | Results* |
|---|---|---|---|---|---|---|---|---|
| Global health status | Time | 2 | 5.50 | >0.05 | T0 (I) T1 (J) | - | - | - |
| | | | | | T1 (I) T2 (J) | - | - | - |
| | | | | | T0 (I) T2 (J) | - | - | - |
| | Treatment | 2 | 16.83 | <0.001 | S (I) CRT (J) | 21.374 | <0.001 | Worse (CRT) |
| Physical function | Time | 2 | 14.64 | 0.001 | T0 (I) T1 (J) | -15.749 | <0.001 | Improved(T1) |
| | | | | | T0 (I) T2 (J) | -14.394 | 0.011 | Improved (T2) |
| Role function | Time | 2 | 5.36 | >0.05 | T0 (I) T1 (J) | - | - | - |
| Emotional function | Time | 2 | 8.82 | 0.012 | T1 (I) T2 (J) | 11.887 | 0.025 | Worse (T2) |
| | Treatment | 2 | 7.96 | 0.019 | S (I) CRT (J) | 23.295 | 0.011 | Worse (CRT) |
| | Time*Treatment | 4 | 26.34 | <0.001 | - | - | <0.05 | - |
| Cognitive function | Time | 2 | 0.62 | >0.05 | T0 (I) T1 (J) | - | - | - |
| | | | | | T1 (I) T2 (J) | - | - | - |
| | Treatment | 2 | 7.23 | 0.027 | S (I) CRT (J) | 21.103 | 0.022 | Worse (CRT) |
| Fatigue | Time | 2 | 148.85 | <0.001 | T0 (I) T1 (J) | 26.572 | <0.001 | Improved (T1) |
| | Treatment | 2 | 964.66 | <0.001 | S (I) CT/RT (J) | -10.606 | 0.010 | Worse (CT/RT) |
| | | | | | S (I) CRT (J) | -25.747 | 0.001 | Worse (CRT) |
| | Time*Treatment | 4 | 439.68 | <0.001 | - | - | <0.05 | - |
| Nausea and vomiting | Time | 2 | 190.15 | <0.001 | T1 (I) T2 (J) | - | - | Maintained |
| | | | | | T0 (I) T2 (J) | 13.180 | 0.002 | Improved (T2) |
| | Treatment | 2 | 336.95 | <0.001 | S (I) CT/RT (J) | - | - | |
| | | | | | S (I) CRT (J) | -13.170 | 0.036 | Worse (CRT) |
| | | | | | CT/RT (I) CRT (J) | -12.808 | 0.040 | Worse (CRT) |
| Pain | Time | 2 | 551.85 | <0.001 | T0 (I) T1 (J) | 31.185 | 0.001 | Improved (T1) |
| | Treatment | 2 | 692.95 | <0.001 | S (I) CRT (J) | -38.626 | 0.001 | Worse (CRT) |
| | | | | | CT/RT (I) CRT (J) | -38.529 | 0.001 | Worse (CRT) |
| Dyspnea | Time | 2 | 31.40 | <0.001 | T1 (I) T2 (J) | - | - | - |
| | Time*Treatment | 4 | 68.46 | <0.001 | - | - | <0.05 | - |
| Insomnia | Time | 2 | 126.57 | <0.001 | T0 (I) T1 (J) | - | - | - |
| | | | | | T1 (I) T2 (J) | 39.465 | 0.005 | Improved (T2) |
| | | | | | T0 (I) T2 (J) | 36.013 | <0.001 | Improved (T2) |
| | Treatment | 2 | 64.90 | <0.001 | S (I) CT/RT (J) | -31.251 | 0.031 | Worse (CRT) |
| | | | | | S (I) CRT (J) | -20.712 | 0.026 | Worse (CRT) |
| Appetite loss | Time | 2 | 398.55 | <0.001 | T0 (I) T1 (J) | 28.384 | 0.003 | Improved (T1) |
| | | | | | T1 (I) T2 (J) | - | - | Maintained |
| Constipation | Time | 2 | 35.636 | <0.001 | T1 (I) T2 (J) | - | - | - |
| Diarrhea | Time | 2 | 67.68 | <0.001 | T0 (I) T2 (J) | - | - | - |
| Financial difficulties | Time | 2 | 290.13 | <0.001 | T1 (I) T2 (J) | - | - | Maintained |
| | Treatment | 2 | 237.04 | <0.001 | S (I) CT/RT (J) | -0.189 | 0.013 | Worse (CT/RT) |
| | | | | | S (I) CRT (J) | -3.271 | 0.027 | Worse (CRT) |
| | | | | | CT/RT (I) CRT (J) | -3.080 | 0.041 | Worse (CRT) |

*(Continued)*

**Table 4.** (Continued)

| Quality of Life domains | Effect | Df[3] | Wald χ2 | p-value | Comparisons (pairwise method) | Mean difference (I-J) | p–value (Bonferroni) | Results* |
|---|---|---|---|---|---|---|---|---|
| | Time*Treatment | 4 | 228.09 | <0.001 | - | - | <0.05 | - |

Treatment: S, surgery alone and/or radiotherapy; CT/RT, chemotherapy or radiotherapy; CRT, chemotherapy + radiotherapy. Data adjusted for age, sex, Body Mass Index and disease stage. df, Degree of freedom.

*For global QoL and function scales, a higher score means better function, and better QoL; for symptom scales, a higher scale means higher symptom burden and worse QoL. p <0.05 was considered significant.

embarrassment and stoma care problems domains improved at T0 to T2. Furthermore, chemoradiotherapy showed worse scores than surgery alone in many domains such as global health status, emotional and cognitive functions, nausea and vomiting, pain, blood or mucus in stools, body image, abdominal and buttock pain, bloating, dry mouth and embarrassment. Chemotherapy and radiotherapy were associated with worse QoL than surgery alone in fatigue, insomnia, financial difficulties, embarrassment and men's sexual interest.

Although some domains showed no significant statistical differences in GEE, some clinically important differences were observed. At least 30% of patients showed worse pain, anxiety, weight, flatulence, embarrassement and dyspareunia at different times (Fig 2). Comparing T0 to T2, for example, many domains showed worsening scores (ranging from 20 to 30% of patients) including global health status, cognitive function, fatigue, abdominal pain, fecal incontinence and women's sexual interest.

Regarding follow-up losses, the differences at baseline between the final sample and those who withdrew from the study were tested (S1 and S2 Tables). As we can see, there were no differences between the general characteristics of patients or EORTC-QLQ-C30 and EORTC-QLQ-CR29 at baseline for patients who remained in the study and those lost to follow-up. There were not differences between patients who underwent curative versus palliative intent surgeries or those with temporary or definitive colostomy, except for the financial difficulty domain, in which patients with temporary ostomy showed worse score.

## Discussion

In the present study, QoL of CRC patients living with a colostomy was evaluated using a specific questionnaire on three occasions after surgery, T0 (0–2 months), T1 (3–5 months) and T2 (6–8 months). QoL improved between T0 to T1 and this improvement was maintained at T1 to T2 in some domains; for example, physical function, appetite loss, urinary frequency and dry mouth. Regarding type of treatment, chemoradiotherapy and chemotherapy or radiotherapy were associated with worse QoL in several domains, such as global health status, emotional and cognitive functions, fatigue, nausea and vomiting, pain, blood or mucus in stools, body image, abdominal and buttock pain, bloating and dry mouth. Finally, even when a significant association between time or treatment exposures was not observed, important clinical differences were observed between times.

The results of QoL studies in patients living with a colostomy are contradictory and differences in sample and study designs may contribute to this, as well as the different time of evaluation. Also, comparing patients with colostomy and ileostomy together [17,24] in the same analysis may affect the results, since the characteristics and symptoms of the ileum are different from the colon. Therefore, it is important to evaluate a specific group, as in this study. In addition, the variety of questionnaires can make it difficult to compare results between studies because the domains assessed may be different. According to a 2012 systematic review, studies

**Table 5. Model effect, comparisons and post hoc test of the variables of quality of life of EORTC QLQ-CR29 with time and treatment factors using Generalized Estimating Equations.**

| Quality of Life domains | Effect | Df[3] | Wald χ2 | p-value | Comparisons (pairwise method) | Mean difference (I-J) | p–value (Bonferroni) | Results* |
|---|---|---|---|---|---|---|---|---|
| Urinary frequency | Time | 2 | 219.24 | <0.001 | T0 (I) T1 (J) | 10.950 | 0.005 | Improved (T1) |
| | | | | | T0 (I) T2 (J) | 10.942 | 0.005 | Improved (T2) |
| Blood or mucus in stools | Time | 2 | 376.73 | <0.001 | T0 (I) T1 (J) | 1.213 | 0.001 | Improved (T1) |
| | Treatment | 2 | 28.08 | <0.001 | S (I) CRT (J) | -0.043 | 0.005 | Worse (CRT) |
| Stool frequency | Time | 2 | 18.14 | <0.001 | T0 (I) T1 (J) | - | - | - |
| Body image | Time | 2 | 0.61 | >0.05 | T0 (I) T2 (J) | - | - | - |
| | Treatment | 2 | 6.36 | 0.042 | S (I) CRT (J) | 34.752 | 0.028 | Worse (CRT) |
| Urinary incontinence | Time | 2 | 32.12 | <0.001 | T0 (I) T2 (J) | - | - | - |
| Dysuria | Time | 2 | 144.82 | <0.001 | T1 (I) T2 (J) | - | - | Maintained |
| Abdominal pain | Time | 2 | 280.68 | <0.001 | T0 (I) T1 (J) | 20.152 | 0.005 | Improved (T1) |
| | Treatment | 2 | 48.35 | <0.001 | S (I) CRT (J) | -13.859 | 0.048 | Worse (CRT) |
| Buttock pain | Time | 2 | 69.80 | <0.001 | T0 (I) T1 (J) | - | - | - |
| | | | | | T0 (I) T2 (J) | - | - | - |
| | Treatment | 2 | 447.44 | <0.001 | S (I) CRT (J) | -14.030 | 0.018 | Worse (CRT) |
| | | | | | CT/RT (I) CRT (J) | -14.003 | 0.018 | Worse (CRT) |
| Bloating | Time | 2 | 187.14 | <0.001 | T1 (I) T2 (J) | - | - | Maintained |
| | Treatment | 2 | 212.20 | <0.001 | S (I) CT/RT (J) | -1.144 | 0.021 | Worse (CT/RT) |
| Dry mouth | Time | 2 | 411.88 | <0.001 | T0 (I) T1 (J) | 49.395 | <0.001 | Improved (T1) |
| | | | | | T1 (I) T2 (J) | - | - | Maintained |
| | | | | | T0 (I) T2 (J) | 49.359 | <0.001 | Improved (T2) |
| | Treatment | 2 | 387.02 | <0.001 | S (I) CRT (J) | -22.906 | 0.041 | Worse (CRT) |
| | Time*Treatment | 4 | 504.61 | <0.001 | - | - | <0.05 | - |
| Hair loss | Time | 2 | 39.22 | <0.001 | T0 (I) T1 (J) | - | - | - |
| Anxiety | Time | 2 | 11.79 | 0.003 | T0 (I) T2 (J) | - | - | - |
| Weight | Time | 2 | 2.55 | >0.05 | T0 (I) T1 (J) | - | - | - |
| Sore skin | Time | 2 | 38.89 | <0.001 | T0 (I) T1 (J) | 0.346 | 0.014 | Improved (T1) |
| | | | | | T0 (I) T2 (J) | - | - | - |
| Embarrassment | Time | 2 | 65.30 | <0.001 | T0 (I) T2 (J) | 3.086 | 0.008 | Improved (T2) |
| | Treatment | 2 | 76.23 | <0.001 | S (I) CT/RT (J) | -3.681 | 0.041 | Worse (CT/RT) |
| | | | | | S (I) CRT (J) | -5.967 | 0.048 | Worse (CRT) |
| Stoma care problems | Time | 2 | 694.68 | <0.001 | T0 (I) T2 (J) | 25.839 | 0.005 | Improved (T2) |
| Sexual interest (men) | Time | 2 | 99.97 | <0.001 | T0 (I) T1 (J) | - | - | - |
| | | | | | T0 (I) T2 (J) | - | - | - |
| | Treatment | 2 | 165.31 | <0.001 | S (I) CT/RT (J) | -27.405 | 0.028 | Worse (S) |
| | | | | | CT/RT (I) CRT (J) | 26.112 | 0.048 | Worse (CRT) |
| Impotence | Time | 2 | 170.86 | <0.001 | T1 (I) T2 (J) | - | - | - |
| Sexual interest (women) | Time | 2 | 55.04 | <0.001 | T1 (I) T2 (J) | - | - | - |
| | | | | | T0 (I) T2 (J) | - | - | - |
| Dyspareunia | Time | 2 | 57.61 | <0.001 | T0 (I) T1 (J) | - | - | - |

Treatment: S, surgery alone and/or radiotherapy; CT/RT, chemotherapy or radiotherapy; CRT, chemotherapy + radiotherapy. Data adjusted for age, sex, Body Mass Index and disease stage. df, Degree of freedom.

*For global QoL and function scales, a higher score means better function, and better QoL; for symptom scales, a higher scale means higher symptom burden and worse QoL. p <0.05 was considered significant.

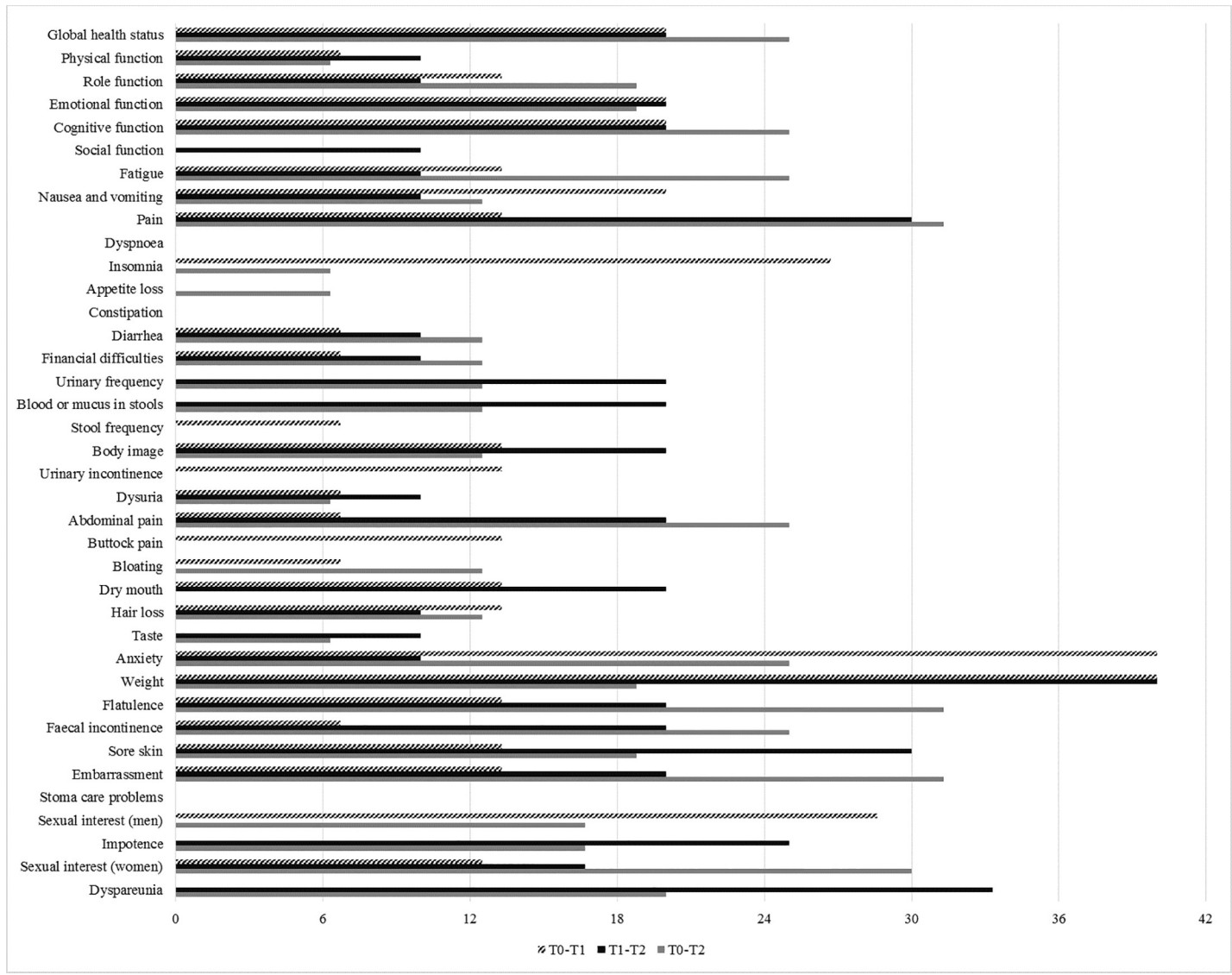

**Fig 2. Percentage of patients worsening of clinically significant difference of EORTC-QLQ-C30 and EORTC-QLQ-CR29 of patients with ostomies due to colorectal cancer at T0 to T1, T1 to T2 and T0 to T2.**

that used general questionnaires tended not to show significant differences, unlike those that used specific instruments [39]. The present study was designed specifically to evaluate only patients living with a colostomy for CRC to better estimate the impact of ostomy and type of treatment on QoL, using a specific questionnaire.

Regarding the function scales, the present study showed an improvement in physical, role and cognitive function at T1 compared to T0 (3–5 months after colostomy); other studies showed a worsening in the first month postoperative compared to preoperative [12,17] and improvement in the sixth month [12,17,18], helping the recovery of health status and minimizing symptoms. Lower scores found on the function scales may be due to surgery and adaptation to the ostomy and daily activities, so the scores increase over time [18]. On the other hand, emotional function showed a worsening at T1 to T2, unlike another study, in which emotional function improved in the sixth postoperative month [17]. Worsening of emotional

function shows that patients need greater psychological support in order to improve their QoL [23].

Abdominal pain, appetite loss and dry mouth were associated with time, improving at T1, probably due to recovery from the surgical procedure. Likewise, pain tends to decrease with tumor resection and postoperatively over time [24]. The domains of nausea and vomiting and stoma care problems showed improvement at T0 to T2. Bowel changes are expected and may occur due to various factors including the size of the intestinal resection and the type of adjuvant therapy; these changes include diarrhea, flatulence, bloating and blood or mucus in stools [15,40]. Therefore, a treatment protocol is usually followed before surgery and once the patient has learned ostomy care, we can see an improvement in these domains over time [5,23].

As expected, treatment may affect the QoL of individuals with colostomies and CRC. In the present study, chemotherapy or radiotherapy, and chemoradiotherapy were associated with later QoL worsening in several function scales and domains. Other studies showed that individuals undergoing neoadjuvant chemoradiotherapy had worse scores for physical, social and role functions [23] and those undergoing adjuvant chemotherapy had worse physical, social, cognitive and emotional functions [5,11]. Our results showed worse scores for emotional and cognitive functions in patients undergoing chemoradiotherapy when compared to those who had surgery alone. Better QoL was expected in patients undergoing surgery alone, as they usually have stage 0 or 1 cancer and surgery is sufficient for complete treatment. Thus, these patients do not suffer as much from the short and long-term adverse reactions of chemotherapy and/or radiotherapy [15].

Chemoradiotherapy was also associated with worse gastrointestinal symptom scores, such as abdominal pain, nausea and vomiting and bloating. Pain, mainly abdominal and perianal, may be one of the most acute toxicities suffered after treatment, [11,19,40,41]. Despite the complications, chemoradiotherapy is usually required for the treatment of individuals with CRC.

Even though there were no significant differences in some domains in the GEE analysis, clinically important difference was found in most. More than 30% of patients showed difference in the pain, anxiety, weight concern, flatulence, embarrassment and dyspareunia domains between times. Anxiety, fears and uncertainties about the future may be present in these patients and psychological monitoring is needed to minimize suffering and facilitate acceptance of the ostomy [6,13,16]. Ostomy implies body changes, and in some situations, bag exposure, leakage or rupture may occur, causing embarrassment [23,42]. In addition, CRC patients can lose weight before surgery and regain it up to the sixth month postoperatively which may contribute to weight concern [12,43].

In the present study, global health status showed no significant difference in GEE analysis. The improvement in QoL scores in the GEE may be due to the resilience and the development of the coping capacity of the patient [19]. Therefore, the worsening of global health status (considering the clinically important differences) may be even greater and often not be noticed by the health team. Some patients tend to rank their current state of health positively compared to the time of diagnosis and surgery. Despite the worsening of some QoL domains, ostomy is a necessary resource and acceptance should be encouraged [16]. Further, other domains such as cognitive function, fatigue, insomnia, abdominal pain, fecal incontinence, women's sexual interest and men's sexual interest also worsened between times. So, clinically important difference should be considered in the health professional's practice, as each patient may experience a worsening of a specific scale or symptom. The health professional team should be aware of any such worsening and consider the patient's QoL; the team is vital in assisting in the process of acceptance, offering care, social, psychological and nutritional support. The multidisciplinary health team and family support are essential for the follow-up and could help with social

reintegration and improvement of disease-specific points, which in turn could assist in the recovery of the patient's QoL [44].

No studies were found showing financial difficulty in patients with temporary or permanent ostomies. The evaluated patients receiving free colostomy bags at the outpatient clinic available by Brazil's publicly funded health care system. This supply was suspended in some moments and this fact could have affected the financial perception in these patients. In addition, a study found showed worsening of some domains in patients undergoing palliative treatment when compared to those undergoing curative treatment, unlike the present study, this may be the fact that the study sample is not specific for colostomy [31].

There were some limitations associated with the study. The first interview was performed up to two months after the ostomy, and a preoperative evaluation would help to better evaluate the QoL. However, all patients submitted to colostomy surgery were invited to participate. Secondly, an interview was used rather than a self-completion questionnaire as many of the patients were elderly and had difficulty understanding the activity. However, all questions were asked exactly as in the questionnaire so as not to induce the answer and we ensured an appropriate response rate for all domains of quality of life. The loss to follow-up was also high but in addition to the size effect and the observation power estimates, we compared baseline characteristics and QoL domains between our sample and those who withdrew from the study, to ensure similarity among them. This showed us that our sample did not suffer a bias at this point. Although patients undergoing palliative treatment could show differences in QoL, since this sample is restricted, the exclusion of these patients could increase the sample losses. Thus, we evaluated curative and palliative intent surgery and difference was not found at the baseline. On the other hand, between temporary and permanent ostomy difference was found in the financial difficulty domain, however all other domains had no difference. Although some domains did not reach enough power, we had adequate sample size and power of observation for several domains. The study also had strengths: the sample consisted only of patients living with a colostomy for CRC, not a heterogeneous sample, which allowed a better evaluation of QoL for this specific clinical condition by time. Thus, the sample is clinically relevant when compared to other studies. In addition, we performed a GEE analysis which is a robust assessment tool in prospective studies adjusting all models by potential confounders.

## Conclusion

Time after of ostomy surgery and type of treatment are associated with changes in QoL in CRC patients living with a colostomy. QoL became better in various domains at 3–5 months and this improvement remained 6–8 months after the colostomy surgery. On the other hand, chemoradiotherapy was associated with a belatedly worse QoL in domains such as nausea and vomiting, bloating and flatulence. Finally, several patients showed worsening clinical difference in several domains, even when this was found to not be statistically significant. Therefore, health teams could use these results to reassure patients that this procedure will improve their QoL in many functional and symptomatic aspects.

## Supporting information

**S1 Table. Differences in the general characteristics of patients who remained in the study and those lost to follow-up.** * Mann-Whitney test for quantitative and chi-square for qualitative variables.
(DOCX)

**S2 Table. EORTC-QLQ-C30 and EORTC-QLQ-CR29 differences between patients who remained in the study and those lost to follow-up.** *Mann-Whitney Test.
(DOCX)

**S3 Table. EORTC-QLQ-C30 and EORTC-QLQ-CR29 differences between patients undergoing curative and palliative surgery and patients with temporary and permanent ostomy.** *Mann-Whitney Test.
(DOCX)

## Author Contributions

**Conceptualization:** Karine de Almeida Silva, Geórgia das Graças Pena.

**Data curation:** Karine de Almeida Silva, Arenamoline Xavier Duarte, Amanda Rodrigues Cruz.

**Formal analysis:** Karine de Almeida Silva, Lúcio Borges de Araújo, Geórgia das Graças Pena.

**Investigation:** Karine de Almeida Silva, Lúcio Borges de Araújo.

**Methodology:** Karine de Almeida Silva, Geórgia das Graças Pena.

**Project administration:** Geórgia das Graças Pena.

**Software:** Karine de Almeida Silva.

**Supervision:** Karine de Almeida Silva, Geórgia das Graças Pena.

**Writing – original draft:** Karine de Almeida Silva, Arenamoline Xavier Duarte, Amanda Rodrigues Cruz.

**Writing – review & editing:** Karine de Almeida Silva, Lúcio Borges de Araújo, Geórgia das Graças Pena.

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
