## [Decision Letter · Decision Letter 0]

14 Jul 2020

PONE-D-20-17764

Time after ostomy surgery and type of treatment are associated with quality of life changes in colostomized colorectal cancer patients

PLOS ONE

Dear Dr. Pena,

Thank you for submitting your manuscript to PLOS ONE. After careful consideration, we feel that it has merit but does not fully meet PLOS ONE’s publication criteria as it currently stands. Therefore, we invite you to submit a revised version of the manuscript that addresses the points raised during the review process.

This is an interesting study that addresses a key patient centered issue. In the revision it will be important to use patient centered language (e.g., "colostomized patient" should be rephrased to "patient living with a colostomy" etc.). Reviewer #1 has raised an important concern that, I too, would like you to expand upon: the follow-up for this longitudinal study was very low, therefore issues to consider include: 1) Why were patients lost to follow-up; did some have ostomy reversal or for some other reason? 2) Were there any factors measured at baseline that were predictive of loss to follow up? And 3) How does your statistical model handle this issue of missing data? It is plausible to think that patients with poor quality of life might withdraw from the study, which would impact your findings if only patients with good quality of life remained in the analytic sample. A robust evaluation and acknowledgement of this issue will strengthen this paper. We look forward to receiving the revised manuscript.

We look forward to receiving your revised manuscript.

Kind regards,

Justin C. Brown

Academic Editor

PLOS ONE

Journal Requirements:

2. Please include a caption for figure 2

Reviewers' comments:

Reviewer's Responses to Questions

**Comments to the Author**

1. Is the manuscript technically sound, and do the data support the conclusions?

Reviewer #1: Yes

Reviewer #2: Partly

2. Has the statistical analysis been performed appropriately and rigorously? 

Reviewer #1: Yes

Reviewer #2: I Don't Know

3. Have the authors made all data underlying the findings in their manuscript fully available?

Reviewer #1: Yes

Reviewer #2: Yes

4. Is the manuscript presented in an intelligible fashion and written in standard English?

Reviewer #1: Yes

Reviewer #2: No

5. Review Comments to the Author

Reviewer #1: I am very honored that the authors and your journal have asked me to review this article. This article is a good study. They used three methods to analyze the recovery of quality of life in patients with colorectal cancer with ostomy, providing guidance for the clinical health treatment team. Can the "using a specific instrument" on line 282 be specific, such as the method or tool used?

Reviewer #2: Overall: nice prospective study with some interesting findings. The English language needs substantial revision with regards to grammar and phrasings. Furthermore, both introduction and discussion sections could benefit from some trimming. References should be scrutinized again; several of the references just mentioned once could be referenced numerous times. Some references I cannot examine owing to language barrier (e.g. 17, 18, 19, 35, 45) and one I cannot find in neither google scholar, pubmed nor embase (#23, Salles et al). I believe the high number of references could be reduced by for example using most recent literature, that of better quality and with those with populations resembling that of the present study.

There is a substantial drop out /loss to follow-up from T0 to T1 and T2. Reason for this should be explained better along with potential significant differences between patients answering T1 and T2 and those who are lost to follow-up. Especially since it is difficult to work out whether this study concerns curative treatment only. If any patients undergo palliative treatment or have recurrence this (curative vs. palliative treatment and recurrence) should be presented in table 1 as it potentially affects HRQoL.

Introduction: Could be trimmed in order to improve presentation of the study objective. First 2/3 seeks to justify the study but a more clear presentation of the problem and study objective would improve the section.

Numerous studies exist on HRQoL in ostomized CRC patients using cancer specific intruments (the EORTC QLQ-C30 and –CR29: Verweij et al 2017, Brauman&Müller et al 2016, Kasparek et al 2011, Mols et al 2014, Zajac 2008, Mahjoubi et al 2012 among others) so this sentence should be rephrased

Methods:

Inclusion/esclusion criteria:It is unclear to me if this is study includes patients undergoing curative therapy alone or if some patients undergo palliative treatment. This is very important when evaluating HRQoL.

It is also unclear to me if some patients have chemotherapy during some (but not all) of time points.

The phrase “Negative clinically changes” can be confusing at times. ‘Improvement’ and ‘worsening’ for example is easier understood.

CR-29 is not exclusively for stoma patients. It is for colorectal cancer patients. This should be changed.

I am not familiar with all of the applied statistical methods, and cannot assess whether appropriate methods are used. There are some repetitions in this sections, and it could be shortened considerably. The last part of the section is especially hard to understand.

Results:

Table 1: As mentioned I would like to see: curative vs. palliative treatment and recurrence in this table. As well as temporary/permanent colostomy distributions.

Tables 2A and 2B: QoL data are skewed but by convention EORTC data are most often presented as means and SD but not necessarily median and IQR. Adhering to this could improve overview of these tables. Or else the reason for presenting both means AND medians should be explained. Significance test should of course consider the skewness of data (non parametric tests)

As mentioned comparison of disease characteristic of patients who answer T1 and T2 and those who are lost to follow-up should be presented. There are a substantial risk of bias here that must be addressed. If you know reasons who patients dropped out these should be presented.

Discussion: This section could also be trimmed considerable focusing on the research questions covered by this study. Some sentences are completely abundant.

It is highly debatable if it makes any sense discussing differences that are non significant even if clinically relevant (since they were significant). And drawing conclusions based on this should be done with more caution.

The discussion of limitations and strengths should be more exhaustive. Using interviews rather than self-completion should be explained. The high number of loss to follow up should be addressed and if all patients are not treated with curative intent this should be stated. This should also the timing of oncological treatment and how this may have affected outcomes. You cannot write the study has a higher response rate with a drop-out rate of 26/41=63%. It seems you are referring to missing items which will be small in interview-studies.

6. PLOS authors have the option to publish the peer review history of their article (what does this mean?). If published, this will include your full peer review and any attached files.

Reviewer #1: **Yes: **Guojun Tong

Reviewer #2: No

---

## [Author Response · Author response to Decision Letter 0]

28 Aug 2020

Please, find attached the the response for reviewers togehter with the tracked and clean manuscripts.

---

## [Editor Report · Decision Letter 1]

2 Sep 2020

Time after ostomy surgery and type of treatment are associated with quality of life changes in colorectal cancer patients with colostomy

PONE-D-20-17764R1

Dear Dr. Pena,

We’re pleased to inform you that your manuscript has been judged scientifically suitable for publication and will be formally accepted for publication once it meets all outstanding technical requirements.

Kind regards,

Justin C. Brown

Academic Editor

PLOS ONE
---

## [Editor Report · Acceptance letter]

23 Nov 2020

PONE-D-20-17764R1 

Time after ostomy surgery and type of treatment are associated with quality of life changes in colorectal cancer patients with colostomy 

Dear Dr. Pena:

I'm pleased to inform you that your manuscript has been deemed suitable for publication in PLOS ONE. Congratulations! Your manuscript is now with our production department. 

Kind regards, 

on behalf of

Dr. Justin C. Brown 

Academic Editor

PLOS ONE